

# CRF: detection of CRISPR arrays using random forest

Kai Wang and Chun Liang

Department of Biology, Miami University, Oxford, OH, USA

## ABSTRACT

CRISPRs (clustered regularly interspaced short palindromic repeats) are particular repeat sequences found in wide range of bacteria and archaea genomes. Several tools are available for detecting CRISPR arrays in the genomes of both domains. Here we developed a new web-based CRISPR detection tool named CRF (CRISPR Finder by Random Forest). Different from other CRISPR detection tools, a random forest classifier was used in CRF to filter out invalid CRISPR arrays from all putative candidates and accordingly enhanced detection accuracy. In CRF, particularly, triplet elements that combine both sequence content and structure information were extracted from CRISPR repeats for classifier training. The classifier achieved high accuracy and sensitivity. Moreover, CRF offers a highly interactive web interface for robust data visualization that is not available among other CRISPR detection tools. After detection, the query sequence, CRISPR array architecture, and the sequences and secondary structures of CRISPR repeats and spacers can be visualized for visual examination and validation. CRF is freely available at http://bioinfolab.miamioh.edu/crf/home.php.

# INTRODUCTION

Found in bacteria and archaea genomes, CRISPRs (clustered regularly interspaced short palindromic repeats) are special DNA sequences that are arrays of direct repeats (23–55 $nt$) separated by different spacers with similar length. The first description of CRISPRs was in 1987 (*Ishino et al., 1987*). Acting as an acquired immune system, CRISPR arrays can embed foreign DNA fragments into spacers and defend host prokaryotic genomes against the invasion of viruses or plasmids (*Barrangou et al., 2007*). Accordingly, some of spacer sequences can perfectly match to viruses or plasmids genomes. CRISPR arrays can be classified into three systems based on the mechanism of inducing adaptive immunity (*Hsu, Lander & Zhang, 2014*). In both Type I and Type III CRISPR systems, the CRISPR RNAs (crRNAs) are generated from cleavage of pre-crRNAs by CRISPR associated ribonucleases. These crRNAs can bind to certain proteins to form RNA-protein complexes that can degenerate invading DNAs. In Type II CRISPR system, a pre-crRNA pairs with a *trans*-activating CRISPR RNA (tracrRNA) to form a RNA-RNA duplex, which can be identified and cleaved by certain nucleases. After cleavage, the resultant crRNA-tracrRNA duplex can bind to Cas9 protein to form a protein complex that can degenerate invading viruses or plasmid DNAs. As one example of Type II system, CRISPR-Cas9 currently is a popular

Corresponding author
Chun Liang, liangc@miamioh.edu

tool for genome editing. With artificial crRNAs, the CRISPR-Cas9 complex can be used to accurately knockout target genes in eukaryotic genomes for genome editing. In order to design efficient crRNA to knock out target genes, detection of CRISPR arrays in bacteria and archaea genomes is the first essential step (*Ran et al., 2013*).

Currently, several tools are available for detecting CRISPR arrays. CRT (*Bland et al., 2007*) was implemented in JAVA and PILER-CR (*Edgar, 2007*) in C++ as executable programs. Meanwhile, three other tools were implemented as web-based tools: CRISPRFinder (*Grissa, Vergnaud & Pourcel, 2007b*), CRISPI (*Rousseau et al., 2009*), and CRISPRDetect (*Biswas et al., 2016*). All these existing programs use either predefined parameters for seed region detection and extension in repeat finding or a scoring system for CRISPR identification. For instance, PILER-CR finds CRISPR candidates through local alignments of a query genome and uses sequence similarity and length requirements as its criteria to filter out some invalid CRISPR arrays (*Edgar, 2007*). Differently, CRT deploys the strategy of finding *k-mer* seed regions and conducting sequence extension in repeat detection. After obtaining all putative candidates, CRT applied simple criteria of sequence similarity and length requirement to exclude invalid CRISPR arrays (*Bland et al., 2007*). In CRISPRFinder (*Grissa, Vergnaud & Pourcel, 2007b*), Vmatch (*Kurtz, 2003*) was used to find potential CRISPR candidates first, then a scoring system was applied to score all candidates, and lastly CRISPR candidates that meet certain score and length requirements were selected as valid ones after eliminating tandem repeats using ClustalW (*Thompson, Higgins & Gibson, 1994*). As the most recent tool for CRISPR detection, CRISPRDetect also utilizes a *k-mer* based seed region detection and extension strategy, scores all candidates using a complex set of criteria, and reports those candidates with high scores as the detected CRISPR arrays (*Biswas et al., 2016*). CRT, PILER-CR and CRISPRFinder are powerful in detecting CRISPR arrays and are very easy to use. However, their filtration steps are not accurate enough to exclude all invalid CRISPR arrays, as demonstrated in our data analysis. Our study also revealed that many CRISPR arrays detected by other tools are missed by CRISPRDetect, and many larger integral CRISPR arrays were split into and reported as smaller CRISPR arrays by CRISPRDetect (see Data S1 ).

Differentiating valid CRISPR arrays from invalid ones is very challenging. CRISPR candidates with canonical repeat architecture might not be real or valid due to other sequence features (e.g., secondary structure properties) that might affect DNA-RNA-protein interaction (*Jackson et al., 2014*). As the first tool in CRISPR detection for archaea and bacterial genomes that utilizes machine learning, we developed a new bioinformatics pipeline named CRF (**C**RISPR Finder by **R**andom **F**orest). Innovatively, we applied a random forest classifier to identify the real CRISPR arrays based on repeat sequence and structural features. Instead of using a scoring system to differentiate real and invalid ones, the classifier was used to exclude invalid CRISPR arrays from all candidates. A total of 32 triplet elements that contain both sequence content and structure information were extracted from both known and candidate CRISPR arrays sequences for training and testing the classifier. The classifier achieved a very high accuracy (94.42%) and sensitivity (93.99%), indicating that our classifier is powerful enough to differentiate valid and invalid CRISPR arrays. In CRF, tandem repeat checking was also applied to filter invalid CRISPR

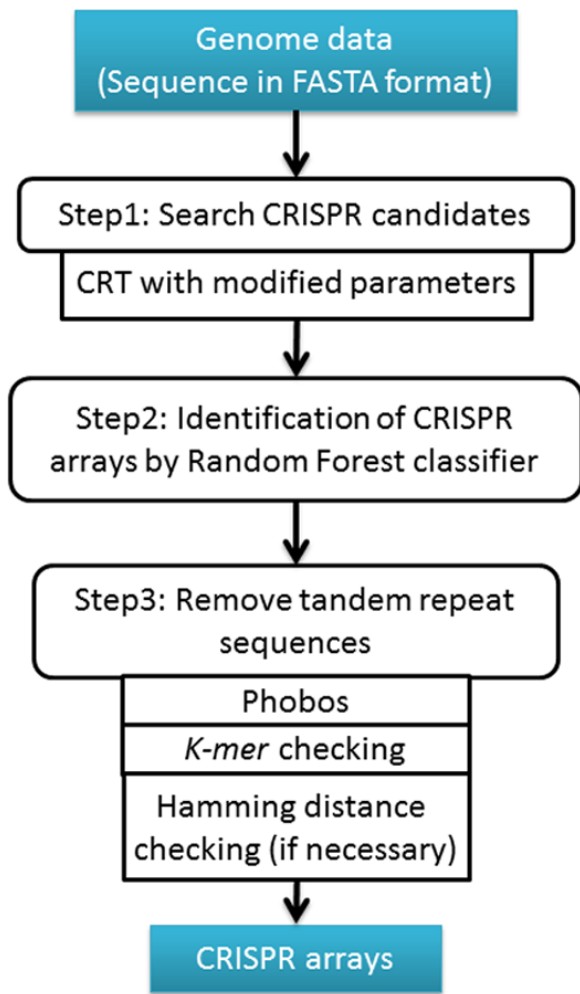

**Figure 1** The workflow of CRISPR Finder by Random Forest (CRF) Pipeline.

candidates. The whole pipeline was implemented in a web-based tool by using PHP and PERL scripts. CRF is able to detect CRISPR arrays and provides a highly interactive data visualization of the query sequence, array architecture, repeat weblogo, and the sequences and secondary structures of both repeats and spacers, etc.

## MATERIALS AND METHODS

CRF contains three parts as shown in Fig. 1. First, CRISPR recognition tool (CRT) was used to detect all CRISPR array candidates. CRT was a widely used tool in finding CRISPR arrays, and showed a better performance than PILER-CR (*Edgar, 2007*). However, the default parameters of CRT are very strict and fail to detect some valid CRISPR array candidates. In order to detect more array candidates, we adjusted the parameters of CRT and particularly changed the seed region of repeats from 8 *nt* to 5 *nt*. Since the repeat sequences of CRISPR arrays could contain mismatches (*Sorek, Lawrence & Wiedenheft, 2013*) and the seed region is first used to search for perfect matches and then extended to
**Table 1** Datasets for random forest classifier training and testing.

| | CRISPR repeats | Random sequences |
|---|---|---|
| Training set | 9,126 | 9,600 |
| Testing set | 2,281 | 2,400 |
| Total number | 11,407 | 12,000 |

tolerate mismatches in repeat detection, a shorter seed region can enhance search sensitivity and allow more CRISPR candidates bearing mismatches detected. For nucleotide extension at the 3′ end of the identical seeds, the default threshold was also changed from 0.75 to 0.6, while the 5′-end extension threshold was the default (i.e., 0.75) in CRT. This means that we can extend the repeat seed region one nucleotide right (3′) if at least 60%, rather than 75%, of repeats share the same nucleotide in this position. The reason why we modified the threshold of 3′ end extension, rather than 5′ end extension, is that the 3′ ends of CRISPR repeats appear to have more variations than their 5′ ends (*Jiang et al., 2013*). Obviously, such a change in the extension threshold allows for detection of more candidate CRISPR arrays.

Second, we trained a random forest classifier to differentiate valid CRISPR repeats from invalid repeat candidates. All CRISPR arrays from CRISPRdb were downloaded and the repeat sequences from these known CRISPR arrays were extracted for training. After removing the redundancy, 11,407 repeats were used as the positive dataset. The negative dataset was the random sequences generated from the positive dataset by using a first-order Markov model. There were 12,000 random sequences within the negative dataset. Sequence content and secondary structure analyses of direct repeats in CRISPR arrays reveal that some CRISPRs have stable stem-loop structures due to the palindromic nature of their repeats (*Kunin, Sorek & Hugenholtz, 2007*). Thus, we extracted the triplet elements of repeat sequences as feature vectors to train, classify and predict repeat sequences. Such triplet elements prove to be very discriminative in identifying short sequences possessing stem-loop structures because these kinds of features contains both sequence content and structure information (*Wang et al., 2014*; *Xue et al., 2005*). As shown in Fig. 2, 32 triplet elements can be used to represent each repeat sequence effectively. In CRF, the secondary structures of repeats were predicted by RNAfold (*Hofacker et al., 1994*). The random forest model was trained using randomForest (version 4.6-10) R package (*Liaw & Wiener, 2002*). The positive and negative datasets were randomly divided into a training set (80%) and a testing set (20%) as shown in Table 1. The random forest classification model was trained with the default parameters (i.e., ntree is 500, mtry is 5, and the threshold of splitting criterion is 50%). The input sequences will be labeled either "repeat" (i.e., CRISPR repeat) or "random" after the classification.

Repeats, spacers, and whole CRISPR arrays need to be checked to exclude potential tandem repeats that are unlikely to be true CRISPR arrays (*Skennerton, Imelfort & Tyson, 2013*). We used Phobos (*Mayer, 2008*) to identify potential tandem repeats among all CRISPR candidates we detected. Phobos gives each candidate a score for being a tandem repeat, and we used the default threshold score of ≥10 for tandem repeat determination. Moreover, a *k-mer* ($k = 3$) check was applied to exclude potential tandem repeats. The

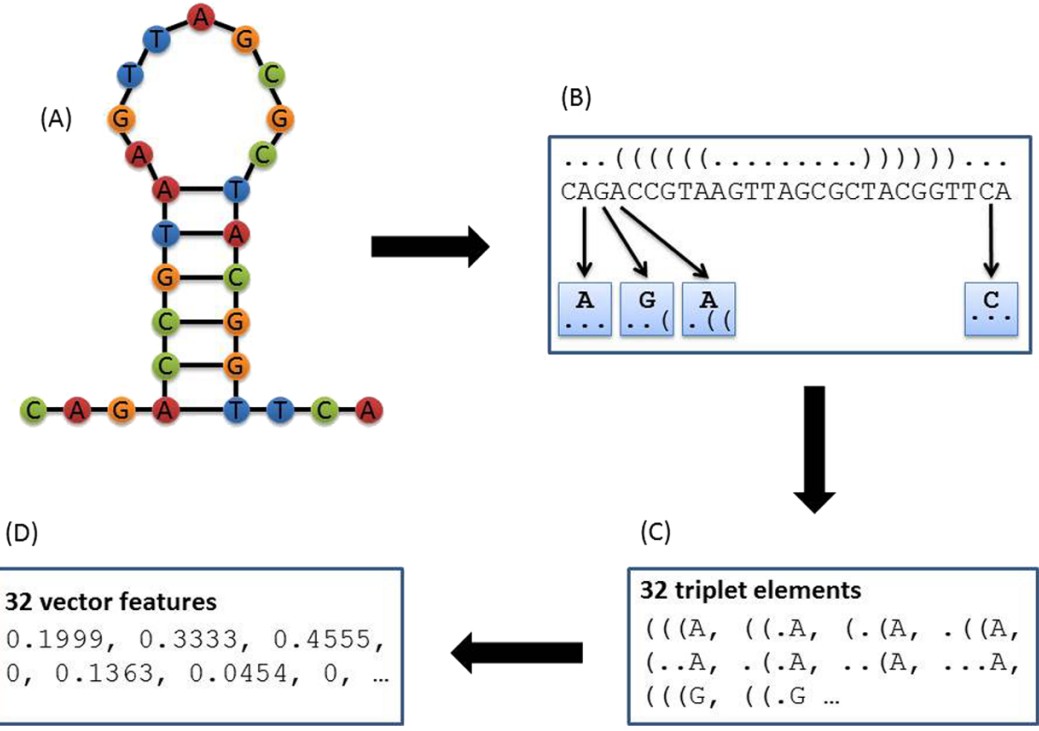

**Figure 2 Principle of triplet elements that contain both sequence and structural information.** The secondary structure was predicted by RNAfold as shown in (A). The triplet elements consist of the dot-bracket structure and the sequence content. (B) A moving window (length = 3) will slide over from the left end to the right end of the dot-bracket structure, and a sub-structure with the length of 3 will be extracted each time. Each sub-structure combined with the middle nucleotide of the three adjacent nucleotides will be counted as one triplet element. (C) Because there are only two structural statuses for each nucleotide (paired/bracket versus unpaired/dot), in total, three adjacent nucleotides will have 8 different structural combinations (e.g., "(((", "((.", "(..", "(.(", ".((", ".(.", "..(", and "…"). With "A", "G", "C", and "T" nucleotides, there are 32 triplet elements that have both sequence content (i.e., nucleotide) and structural information. So for each sequence, the number of 32 triplet elements will be counted and normalized through dividing by the total element number. The normalized data was shown in (D).

default *k-mer* (*k* = 3) threshold values for repeats, spacers and whole CRISPR arrays are 0.2, 0.2, and 0.25 respectively. These threshold values were defined based on the known CRISPR arrays from CRISPRdb. Users can modify these values through "Advance Settings" on the home page of CRF to get more specific results. Furthermore, in order to exclude invalid CRISPR arrays, Hamming distance was calculated to see if a CRISPR array contains same-length spacers. Among all spacers in the CRISPR array, if any two spacers have a similarity larger than 50%, then the CRISPR array candidate will be discarded (*Barrangou & Marraffini, 2014*).

## RESULTS

### Random forest classifier

We used a random forest classifier to differentiate real CRISPR repeats from invalid ones using triplet elements as sequences features. After the training and testing processes, the model achieved accuracy (94.42%) and sensitivity (93.99%). This high accuracy indicates

that our classifier is able to differentiate real CRISPR repeats from invalid CRISPR repeats. The **A**rea **U**nder **C**urve (AUC) of this classifier is 0.9837. This high AUC value suggests that this classifier has great performance in CRISPR repeat identification. Each CRISPR array candidate will be classified as a valid or invalid CRISPR array based on their repeat sequences.

## CRF web service

The web interface of CRF was implemented in PHP and JavaScript whereas PERL was used for its backend pipeline. The web service consists of three pages ("*Home*", "*Manual*", and "*Contact*"). The query sequence should be one in FASTA format and ambiguous nucleotide N characters are accepted. Users can use the default settings or click the "*Advanced Settings*" button to modify parameters for CRF. These parameters are mainly about changing the sequence length of repeats and spacers and adjusting values to remove tandem repeats from the CRISPR candidates detected.

After submitting a query sequence on the CRF website, the final valid CRISPR arrays are shown in a summary table. On the top of the results page, the query sequence can be displayed and visualized in SeqViewer by clicking the "*Show sequence in SeqViewer*" button. In SeqViewer, users can search sequence motifs by using the motif search input field on the navigation bar. The use of IUPAC codes is acceptable for the motif search function, which enables powerful fuzzy search of motifs. For the detailed results of each detected CRISPR array, a user can click the ID/Name on the result list to jump to the relevant position on the result page. The detailed results of each CRISPR array consists of four parts: (1) the basic information of a CRISPR array is listed at the beginning of the detailed results, (2) a SVG-based CRISPR array architecture graph and the sequence follows as the second part, where a user can click the sequence to show secondary structure of each repeat or spacer, (3) the sequence logo figure of CRISPR repeats obtained through WebLogo 3.4 (*Crooks et al., 2004*) is displayed next, and (4) the web link that allows a user to download relevant figures and data is provided last. Users can download the spacers or the whole CRISPR array as a sequence file in FASTA format.

## Comparison with other tools

For performance comparison, we compared CRF with PILER-CR, CRISPRDetect and CRT, we only choose these three programs for comparison because CRISPRFinder and CRISPI are web-based tool that do not offer source codes for local machine installation. We tested these four different programs using the same genome dataset that has been utilized for random forest classifier testing. The non-redundant repeat sequences reported by CRISPRdb that were used as the testing dataset in our study were extracted from 1,139 genomes. These 1,139 genomes were used for comparison. All four programs were executed with their default parameters and settings. The results were compared to CRISPRdb, because CRISPRdb is the most comprehensive, widely utilized CRISPR database (*Grissa, Vergnaud & Pourcel, 2007a*). Based on the detected CRISPR arrays, we found that PLIER-CR can detect more arrays that begin or end at wrong positions in comparison with the known CRISPRs from CRISPRdb (*Bland et al., 2007*). CRISPRDetect showed better results, but it
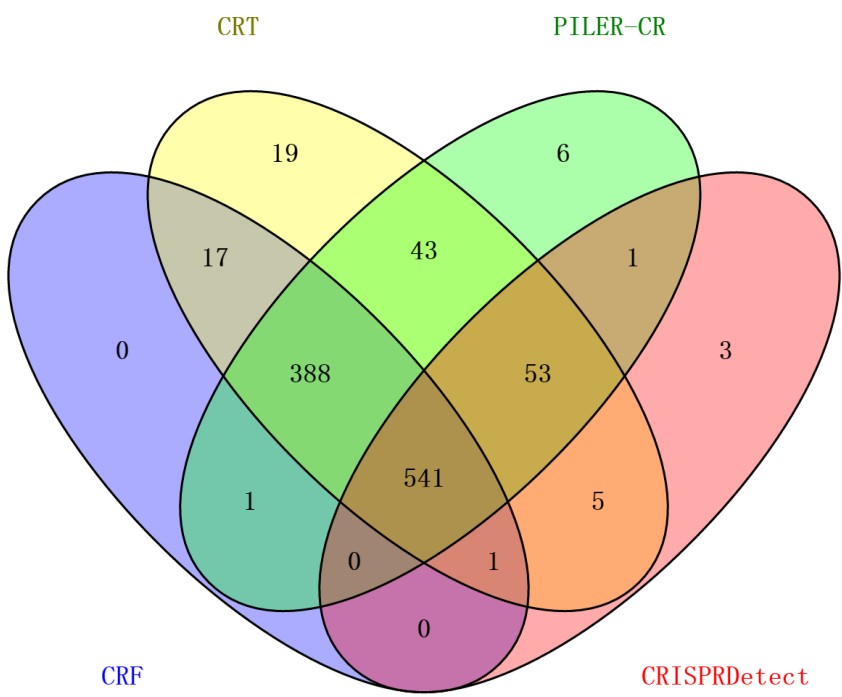

**Figure 3** Comparison of the number of genomes detected with CRISPR arrays by four programs.

also missed certain portions of CRISPR arrays. CRT is better than other the two tools, but it cannot filter out tandem repeat sequences.

The result shows that 541 out of 1,139 genomes can be detected with CRISPR arrays by all of these four programs (Fig. 3). Individually, PILER-CR, CRT, CRISPRDetect and CRF detected CRISPR arrays in 1,033, 1,067, 604 and 948 out of 1,139 genomes, respectively. In CRISPRdb, those arrays with only one spacer flanked by two direct repeats are also maintained as "questionable structures" (*Bland et al., 2007*), whereas a minimum of two spacers with three direct repeats are required for being valid CRISPR arrays by all four programs. This explains why all four programs detected few genomes with CRISPR arrays. Comparing these four programs, PILER-CR, CRT and CRF detected CRISPR arrays in 80% of 1,139 genomes whereas CRISPRDetect detection rate was only 50%. Clearly, this indicates that CRISPRDetect missed a significant portion of CRISPR arrays, in comparison with the other three programs.

In CRISPRdb, 3,689 CRISPR arrays are distributed in 1,139 testing genomes. We used these CRISPR arrays as references to calculate sensitivity and precision for these four programs. The results are shown in Table 2. CRF, CRT, and PILER-CR achieved above 60% sensitivity (*i.e.* CRF: 61.6%, CRT: 66.2%, and PILER-CR: 69.7%). CRISPRDetect only achieved 36.5% sensitivity, and this indicates that CRISPRDetect missed almost half of the CRISPR arrays in these testing genomes. For precision, CRF achieved 83.3% while the other programs only achieved 73.8% (CRT), 51.9% (CRISPRDetect), and 75.2% (PILER-CR). This result suggests that our program is more effective in excluding fake CRISPR arrays than other programs.

**Table 2** Comparison of sensitivity and precision between four programs.

|  | CRF | CRT | CRISPRDetect | PILER-CR |
|---|---|---|---|---|
| Sensitivity (%) | 61.62 | 66.28 | 34.46 | 69.75 |
| Precision (%) | **83.29** | 73.80 | 51.93 | 75.21 |

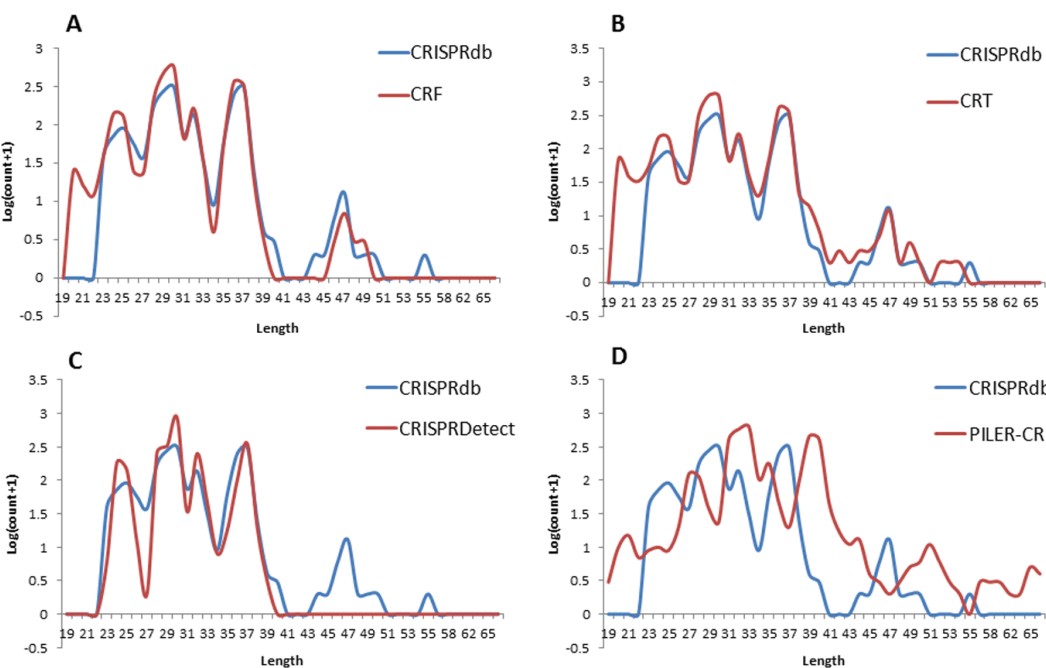

**Figure 4** Repeat length distribution of CRISPR repeats detected by four programs, in comparison with the complete CRISPR repeats annotated in CRISPRdb.

CRISPR repeats are defined as short palindromic repeats, and the length and palindromic structure of the repeats are crucial for interaction between CRISPR RNAs and Cas proteins (*Nishimasu et al., 2014*). However, the repeats do not show an obvious palindromic structure in Type II and Type III CRISPR systems (*Hsu, Lander & Zhang, 2014*). Therefore, we focused on the length distribution of the repeats from detected CRISPR arrays to compare the four programs. The CRISPR repeats from the CRISPR arrays detected from the test data set by all four programs were extracted for length distribution comparison. As shown in Fig. 4, it shows that CRT, CRISPRDetect and CRF have a similar distribution pattern located in repeat length ranged from 23 *nt* to 40 *nt*, which is consistent with that of CRISPRdb. In CRISPRdb, because a strict cutoff (minimum repeat length equals 23 *nt*) was applied, no repeat shows a length in the range from 20 to 23 *nt*. In our CRF pipeline, we retained the CRISPR arrays that have repeat lengths located in the range of 20–23 *nt*, since CRISPR arrays with repeat lengths of 20–23 *nt* have been reported by *Jansen et al. (2002)*. The CRISPR repeats from PILER-CR show a different distribution pattern in which the repeats are longer than the repeats detected by the other three programs. Although PILER-CR can detected more CRISPR arrays than other three programs, Fig. 4 supports the finding that CRISPR arrays from PILER-CR are more likely to start or end at wrong

positions in comparison with CRT and PATSCAN (*Bland et al., 2007*). In terms of length distribution, CRT, CRISPRDetect and CRF show better results PILER-CR. However, CRT cannot filter out the tandem repeat sequences in its reported CRISPR arrays (see Supplemental Information 1). In our CRF pipeline, Phobos was used to remove tandem repeats candidates. For longer repeats ranging from 44 to 50 *nt*, CRISPRDetect detected nothing in this range while CRF, CRISPRdb, and CRT showed a small peak value of around 47 *nt*. Considering the fact that only 604 genomes were detected with CRISPR arrays by CRISPRDetect, it is clear that that CRISPRDetect still fails to detect a noticeable portion of CRISPR arrays. Our data analysis suggests that CRF is a better program for CRISPR array detection due to its consistency with CRISPRdb regarding repeat length distribution, its novelty in applying machine learning to reduce invalid candidates, and its utilization of an effective strategy to filter out tandem repeats.

## CONCLUSIONS

Currently, CRISPR-Cas system is one of the most popular genome editing tools. In order to design effective CRISPR RNA, detecting CRISPR arrays from bacterial and archaea genomes is an important step. Several tools are available for detecting CRISPR arrays. As a valuable addition, we designed a new pipeline called CRF to detect CRISPR arrays from genome data, in which we utilized machine learning and tandem repeat filters. Using 1,139 genomes, we compared CRF with other three tools CRT, PILER-CR, and CRISPRDetect. Clearly, CRISPRDetect failed to detect a significant portion of CRISPR arrays, PILER-CR found more CRISPR arrays at wrong end or start positions, and CRT cannot filter out tandem repeat sequences. Our comparison indicates that CRF shows better performance than the other programs. Moreover, differently from the other tools, CRF offers a highly interactive web interface useful for data visualization, sequence examination and CRISPR architecture validation.

## ACKNOWLEDGEMENTS

We thank Matthew Spinelli and Joshua Hoeksema for comments and suggestions to improve this manuscript.

### Funding

This project is funded by the Committee on Faculty Research (CRF) Program, the Office for the Advancement of Research & Scholarship (OARS), and the Department of Biology, Miami University, Oxford, Ohio, USA. The funders had no role in study design, data collection and analysis, decision to publish, or preparation of the manuscript.

### Grant Disclosures

The following grant information was disclosed by the authors:
Committee on Faculty Research (CRF) Program.

Office for the Advancement of Research & Scholarship (OARS).
Department of Biology, Miami University, Oxford, Ohio, USA.

## Competing Interests

Chun Liang is an Academic Editor for PeerJ.

## Author Contributions

- Kai Wang conceived and designed the experiments, performed the experiments, analyzed the data, contributed reagents/materials/analysis tools, wrote the paper, prepared figures and/or tables, reviewed drafts of the paper.
- Chun Liang conceived and designed the experiments, wrote the paper, reviewed drafts of the paper.

## Data Availability

The raw data has been supplied as Supplemental Files.

## Supplemental Information

Supplemental information for this article can be found online at http://dx.doi.org/10.7717/peerj.3219#supplemental-information.

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
