# Peer review of "CRF: detection of CRISPR arrays using random forest"

_PeerJ, doi:10.7717/peerj.3219_

## Round 0.1 · original submission · Major Revisions

Dear Dr Liang,

We cannot publish your article at present. However, if you addressed the concerns related to statistics (Referee 1) and training/testing set (Referee 2), we might reconsider our decision.

Sincerely
G. Tartaglia

Reviewer 1 ·

Basic reporting

The English language is clear and professional. Minor comments:
- Line 139 - CRISPR is misspelled.
- Lines 147-149 - It should be separated into two sentences.

The introduction on CRISPRs is not completely satisfying. The authors should emphasise the relevance of such elements and why it is important to identify them.

Figure 3 is not useful as it is. It should include the ROC curves of the other classifiers/methods.
Figure 4-6 are very difficult to interpret. The authors should find another way to represent the results.

No code or datasets (training and testing) are provided for the method.

Experimental design

As stated above, it is not clear the impact of the CRISPRs identification and, therefore, it is difficult to establish the relevance of these results. The authors are not reporting the ROC curves (or sensitivity/specificity values) for the other available tools, thus hindering the evaluation of the results in a broader context.
There is no description of the random forest algorithm (e.g., coding language, package, reference, etc.), the options it was run with (e.g., number of trees in the forest, etc.) and how these were chosen (e.g., cross-validation with different hyper-parameter sets, etc.).
It is not specified what is the output variable (continuous or discrete, categorisation or regression tree) and the splitting criterion. Have the authors explored other machine learning approaches before coming to this one?

Validity of the findings

The method was trained using the CRISPRdb data. Therefore, it is not surprising it performs better when repeats from CRISPRs are used as reference. The comparison should be carried out on a set of sequences that were not present within the training dataset, or released on CRISPRdb after the initial retrieval of the training set.

It would be useful to have a comparison of the identified CRISPRs across the tested methods (e.g., Venn diagram), instead of reporting the ‘Uniquely identified CRISPR arrays by each program
in comparison to CRF’.

The conclusion should be expanded , summarising the results and highlighting the novelty or better performance of the method compared to available alternatives.

Additional comments

Although the paper is well written, the method here presented seems a mere extension of an already available tool, namely CRT. The random forest algorithm introduced in this new approach is not sufficiently described. Moreover, the comparison with other available methods is not presented in a clear way, thus, hampering the correct evaluation of CRF performance with respect to its alternatives.

Reviewer 2 ·

Basic reporting

- The text is acceptable, there are some places needing grammar fixes.
- Line 12 – not relevant to include names of other tools in abstract
- Line 48 – “those good CRISPR candidates” bad grammar
- Multiple places where CRISPR is misspelled
- Line 126 – Remove sentence starting with “The whole pipeline…”
- Should give background on who wants to use this tool and why. Sure, CRISPRs are useful immune systems, but why does anyone need to find it in x genome ?
- Instead of Table 2, a venn diagram of the numbers would be easier to digest.
- See comments below about figures 3-6

Experimental design

- As noted above, there should be explanation what this type of tool is for
- Line 39 – Why are predefined parameters for seed region detection and extension not good? Will a biologist using this tool know what parameters to use?
- Line 80 – its not clear what these parameters are and why those changes are better. Maybe just a high level explanation of why those changes increase sensitivity
- It’s necessary to show the performance of CRF compared with the other existing methods. Of course, you’ll need to restrict the testing data to the same ones in your validation set.
- On the same note, Fig 3 should be showing the AUCs for each of the methods. One curve for CRF is not very informative. Consider using precision recall curves as well.
- Possible circularity: are the genomes in your training set overlapping with the genomes in the “performance comparison” set? Not only would that be an unfair comparison with the other methods, you are just testing your method with what you just trained it on…
- You describe method x finds this many, overlaps this amount with method y, etc. Can you dive deeper and perhaps find characteristics of the types of sequences certain methods are finding? Are any of the unique sequences valid? Are they more likely to be valid if more than one method identifies it? Again, some plots comparing FP, FN, TP, TN rates would be helpful.
- The whole section on repeat comparisons starting at Line 168 is confusing. Please explain the relevance of the length of the repeat. Why does the length matter? Are some lengths known to be invalid? There is a lot of time spent describing what the lines on the plot look like, but I don’t know how to interpret it. Furthermore, they don’t look that different to me—jagged lines with a big hump round 30 and a smaller hump around 47.

Validity of the findings

- See a concern about circularity above
- The conclusion is currently just describing CRF is a tool. It should include some statement on how it improves upon existing tools in some way.

---

## Round 0.2 · Major Revisions

Due to lack of details on the testing performances of the models we cannot accept the paper in its present form. If you are able to revise it to respond to the concerns of Reviewer 2 then we can consider if again.

Reviewer 2 ·

Basic reporting

Language was clear and professional.

Experimental design

I don’t understand where 1139 genomes comes from, the table shows the number in the different sets and none are 1139.

I appreciate the longer explanation about length, but I am still unclear on how this affects the result. If you want to show the importance of using Length as a feature, perhaps demonstrate AUC/PrecisionRecall/Some performance measure with and without using length, or using different lengths. Can you just ignore the results of the “wrong” length from another method and get the same result?

I have a hard time interpreting he new Fig 4. I cannot differentiate the two blue lines and there are many lines on top of each other. Might it be easier to if they are each on a separate axis? Am I supposed to see that CRF follows the shape of CRISPRdb the best?

Validity of the findings

I was disappointed to find that instead of adding the ROC plots for the 3 existing methods, the authors chose to completely remove the figure. I previously stated, “It’s necessary to show the performance of CRF compared with the other existing methods”.

The authors go on to report that “the model achieved ~93% accuracy and sensitivity.” Did it achieve 93% accuracy and 93% sensitivity? These are two different measures and they happen to be the same?

I don’t follow the logic that because one method detects more CRISPRs, the other method is clearing missing some. The results of a method can be composed of any combination of TPs, TNs, FPs, and FNs. That is why it is important to show ROC and/or Precision Recall when you are comparing methods.

---

## Round 0.3 · accepted · Accept

I think that the work improved substantially.